# Out Like a Light: Feasibility and Acceptability Study of an Audio-Based Sleep Aide for Improving Parent–Child Sleep Health

**DOI:** 10.3390/ijerph19159416

**Published:** 2022-08-01

**Authors:** Alicia Chung, Peng Jin, Dimitra Kamboukos, Rebecca Robbins, Judite Blanc, Girardin Jean-Louis, Azizi Seixas

**Affiliations:** 1Department of Population Health, NYU Grossman School of Medicine, New York, NY 10016, USA; pj691@nyu.edu (P.J.); demy.kamboukos@nyulangone.org (D.K.); 2Division of Sleep Medicine, Harvard Medical School, Boston, MA 02115, USA; rrobbins4@bwh.harvard.edu; 3Department of Psychiatry and Behavioral Science, University of Miami Miller School of Medicine, Miami, FL 33136, USA; jxb5231@miami.edu (J.B.); girardin.jean-louis@miami.edu (G.J.-L.); azizi.seixas@med.miami.edu (A.S.)

**Keywords:** sleep, child, family, audio story, m-health

## Abstract

Our study examines the acceptability and feasibility of Moshi, an audio-based mobile application, among children 3–8 years old using a parent–child dyadic approach. Our 10-day within-subject pre–post study design consisted of five nights of a normal bedtime routine and a subsequent five nights exposed to one story on the Moshi application during the intervention. Each five-night period spanned three weeknights and two weekend nights. The Short-Form Children’s Sleep Habits Questionnaire (SF-CSHQ) was used to measure children’s sleep at baseline and post-intervention. The PROMIS, Epworth Sleepiness Scale and Pittsburgh Sleep Quality Index were used to assess parents’ sleep. Among the 25 child–parent dyads, the mean child age was 4 (SD = 1.23) and 63% were male (*n* = 15). Mean parent age was 35 (SD = 5.83), 84% were female (*n* = 21), and 48.0% were Black (*n* = 12). For child-only comparisons, mean post-SF-CSHQ measures were lower compared to baseline. A trend in parent sleep is reported. This study shows the potential of an audio-based mobile sleep aid to improve sleep health in a racially diverse parent and child dyad sample.

## 1. Introduction

Sleep timing, duration, habits and quality are essential for a child’s physical, cognitive, and emotional development [1]. Research shows that sleep problems have a high prevalence throughout childhood and adolescence, with 25% to 50% of preschoolers experiencing sleep-related issues [2]. Poor sleep health during early childhood impacts health outcomes, cognitive performance and emotional wellbeing into adolescence [3,4]. Both the American Academy of Sleep Medicine and National Sleep Foundation advise that children 4–12 months old should receive 12–16 h of sleep, children 1–2 years old should receive 11–14 h, children 3 to 5 years old should receive 10 to 13 h, and children 6–13 years-old should receive 9–11 h of sleep [5]. However, several exogenous (e.g., artificial light exposure) and endogenous factors (i.e., chronotype) contribute to difficulties going to bed, issues with falling and staying asleep, intermittent waking episodes, night waking, and waking up too early—all of which can lead to sleep disturbance, and shorter total sleep time.

Night wakings are the most common sleep problem in infants, toddlers and preschoolers, affecting up to 50% of children over six months [4]. These sleep disturbances can be problematic for both children and parents. While sleep disturbances occur throughout the life-course, they are typically experienced during critical developmental stages. Infancy and toddler years both mark significant shifts in sleep and circadian rhythm patterns and behaviors, leading to significant sleep disturbance. Although these shifts in sleep and circadian rhythms generally normalize by early childhood, exposure to destabilizing and disruptive environmental (e.g., noise, light) [6,7] and behavioral (e.g., evening routine) factors can affect the establishment of normalized sleep patterns [8].

### 1.1. Environmental Factors That Affect Bedtime and Sleep Routine

Light and noise are key aspects of the ambient home environment that impact the bedtime and sleep routines of children [9]. Specifically, exposure to artificial light before bedtime hours suppresses the release of melatonin, a sleep-promoting hormone that typically increases hours before bedtime. Acute and extreme exposure to artificial light before bedtime may delay the circadian phase of the melatonin rhythm [10,11], thus delaying the onset of sleep [12]. Melatonin suppression by a light stimulus of 580 lux (typical indoor light levels) in children has been reported to be almost twice that of adults [13]. Artificial light exposure that interferes with the body’s endogenous sleep circadian rhythm may affect a child’s natural sleep pattern and duration. Artificial light at night is also associated with increased perceived stress among young adolescent children [14] and adversely affects circadian timing, sleep physiology, and alertness in children as young as 6 years old [12,15]. The Moshi app proposed in the intervention is a substitute for screen-time devices, as an audio-based mobile application.

Audio-based music therapy and storytelling have emerged as promising solutions to ameliorate environmental noise challenges, without adding to artificial light exposure. External noise pollution from traffic or household noise both have damaging consequences on sleep duration and quality [16,17]. Mitigating the disruptive effects of environmental and behavioral cues on sleep requires a solution that provides a non-stimulating sleep environment. White noise therapy, stochastic noise, and pink noise, with a gradual softness of volume, have been recognized as promising strategies to inculcate sleep onset and maintenance in children and adults [18,19,20]. Music therapy and storytelling are accessible, economical and effective solutions for creating a relaxing environment that promotes healthy sleep and manages sleep disturbance, night wakings, and insomnia in children and adults [21,22]. Music therapy is recognized as being effective for reducing stress and improving mood, key constructs that aid in calming the mind and cue the body to sleep [23], in home and hospital sleep environments. In a preliminary pilot study with 31 infants and school-aged children assigned to a 3-day music therapy or storytelling intervention group while being hospitalized for an infectious disease, significant improvements were reported in the average number of sleep disturbances [21]. The audio-based feature of the Moshi app intervention includes music therapy and storytelling as a promising strategy to promote sleep health in children. The Moshi app houses bedtime story content, and users can download the app on their phone and choose which sleep tracks they want.

Parenting behaviors around bedtime routines have been widely recognized for promoting healthy sleep and wellbeing for the child and family unit [8]. Some common factors that affect sleep scheduling and sleep habits include family mealtime/feeding practices, hygiene/brushing teeth, literacy/reading books, lullabies, familial stress and parent–child interactions [1,8]. Language-based routines during bedtime—such as storytelling, prayers, singing and reading—may have lasting positive effects for a child’s cognitive development, sleep duration and behavioral problems (i.e., anxiety/aggression). Language-based nighttime routines have reported an inverse association with behavioral problems (i.e., anxiety and aggression) and a positive association with verbal test scores, although they do not fully mediate sleep duration [24]. However, the impact of music therapy or storytelling on sleep outcomes in young children has not been reported. Additionally, given the impact of sleep disturbance on both child and parent wellbeing, an examination of language-based music therapy to aid sleep health in the parent–child dyad is a gap in the literature.

### 1.2. Mobile Health Sleep Aides

Sleep problems in children as young as 4 years old are associated with increased physical, behavioral and emotional problems into adolescence [25,26]. These mental health outcomes are especially detrimental for children who are at a critical stage of their brain development. Sleep aides that can be easily integrated into bedtime routines and support sleep hygiene practices, such as audio-based mood setting or storytelling aids, may ameliorate challenges with a child’s sleep health, reduce parent burden, and ease the transition phase to sleep for both the parent and child.

Nearly 25% of preschoolers are affected by bedtime problems and frequent night wakings, known as the behavioral insomnia of childhood [4]. Audio-based mobile aids that do not require the use of screen viewing may offer calming music to cue the home environment and calm the mood for sleep onset. Moshi was developed by the MindCandy company in London, England. The mobile app delivers sleep stories and music. To this end, our objective is to assess the utility of the Moshi mobile app sleep aid to help parents with sleep onset, duration and habits for children 4–8 years old and their parents. 

The theoretical framework guiding this study is a multi-level engagement framework that considers three domains of factors (cognitive knowledge about the sleep environment, affect/relationship, and behavioral/communication) in strengthening family health behaviors, adapted from Huang et al., 2018 [27]. This framework serves as a multi-level family-engagement approach that considers interconnected social–environmental factors and the effects of family’s routines that influence children’s sleep health behaviors in the parent–child dynamic (see Figure 1). Figure 1 illustrates that children’s health behaviors are influenced by interpersonal relationships, starting with their parents [28], highlighting that parental engagement is essential. Parenting behaviors are often influenced by one’s knowledge about the environmental factors that impact sleep, as well as affect and communication strategies (i.e., mood, tone and word choice) between parent and child. Our framework recognizes that parent behavioral practices and routines, coupled with intrinsic child chronotypes for certain sleep timing that may be biologically inherent (i.e., night owl or early bird) [29], are both key factors influencing a child’s sleep. Understanding how these factors work together is critical for setting the stage for a child’s physical health behaviors during early childhood. Additionally, culture and family structures may influence parenting practices (i.e., setting boundaries for bedtimes) that are critical to shaping a child’s sleep patterns [30]. The home environment level (i.e., ambient music/light) [31] is also a key determinant of a child’s sleep [1,32,33]. This framework holds the intersection of the bio-behavioral model with emotional affect (i.e., mood), parenting practices (i.e., routines) and environmental factors (i.e., noise) that collectively act upon sleep health determinants. Engaging parent–child dyads within the aforementioned three domains may yield bi-directional sleep health benefits for the family unit.

This study fills a gap in the literature by being the first to examine the utility of an audio-based, storytelling mobile application to improve sleep health outcomes in parent–child dyads. The purpose of this study is to examine the feasibility and acceptability of a mobile application designed to be easily integrated into a bedtime routine to improve sleep health among children aged 3–8 years old using a parent–child dyadic approach. We examined the role of the Moshi mobile application on parent–child sleep health duration, onset and night wakings. The study hypothesizes that the children and parents who use a soothing bedtime smartphone app will experience an improvement in sleep. Specifically, (1) children will demonstrate better sleep duration, reduced night wakings, and improved sleep onset at post-intervention than pre-intervention, and (2) parents will demonstrate an increase in sleep quality post-intervention compared to pre-intervention. 

## 2. Materials and Methods

### 2.1. Participants

The current study compares sleep habits and duration in a within-subject crossover study design, whereby each participant serves as their own control. The study period included 5 days (3 weekdays and 2 weekend days) without the intervention (children engaged in their normal sleep routine), and 5 days with the intervention (children were exposed to the intervention). IRB approval was obtained in July 2019, and written informed consent was obtained from participants prior to the beginning of the study.

A total of 20 child–parent dyads were included in the study. The mean child age was 4.29 (SD = 1.23) and 62.5% of the children were male (*n* = 15). For parents, the mean age was 35.9 (SD = 5.83), 16% were male (*n* = 4) and 48.0% self-identified as Black (*n* = 12), while 24.0% self-identified as White (*n* = 6). A total of 76% of the parents were married (*n* = 19), 64% had a bachelor’s degree or higher (*n* = 16), and 100% were employed (*n* = 20), with a median annual household income range of USD 80,000 to USD 100,000. (Table 1).

### 2.2. Inclusion Criteria

Eligible participants were parents aged 18 or older with a child aged 4–8 years old. 

### 2.3. Exclusion Criteria

Parents with a history of psychiatric illnesses or psychiatric disorders and those who were unaware of specific psychiatric diagnoses but had a history of treatment with antidepressants, neuroleptic medications, or major tranquilizers were excluded from the study. Parents who were not present in the household during nighttime hours were excluded from the study. Children who shared a bed with a sibling or another parent/caregiver were excluded. Parents who did not agree to the Moshi privacy terms could not participate in the study.

### 2.4. Procedures

Parents were recruited via word-of-mouth snowball referrals from daycare center directors, elementary school leaders, teachers, parents and administrators. Parents were screened for eligibility in person or over the phone with the pre-screening questionnaire, and provided verbal consent over the phone or in person for the study. Eligible participants were scheduled for an appointment with the study team to be invited to learn about the study and complete baseline surveys. Parents who participated received a USD 100 gift card. 

Each participant (parent and child) served as their own control in the cross-over design by completing subjective measures of sleep based on 5 days of normal sleep routines and 5 days of exposure to the mobile application. All participants recorded their sleep patterns, including the time the child and parent went to bed, sleep onset, wake-up times, total sleep time, and sleep duration for the length of the intervention (10 days) using a sleep diary. Both 5-night periods of sleep monitoring included 3 weeknights and 2 weekend nights. The study included 5 days of initial sleep monitoring (Days 1–5), followed by 5 days of using the Moshi app while simultaneously continuing to monitor sleep (Days 5–10). The 3 weekdays and 2 weekend days were selected based on what was most convenient for the parents. Parents were advised to select these days in sequence, such as a Saturday through Wednesday or Wednesday through Sunday, that would work best for them and their child. After the first week of normal sleep procedures, parents were guided on how to download the app, navigate through it, and become familiarized with it, before starting the 5-day intervention period. Although the two periods of the initial and intervention periods may not have been the same for all participants, we do not anticipate that this would affect the results in any way.

Parents selected 1 story per night of roughly 15–25 min in length for the child participant. Parents also completed measures of sleep for themselves and their child at baseline and at the end of the intervention. NYU Grossman School of Medicine IRB study approval was obtained in August 2019. Recruitment and data collection occurred from September 2019 through April 2020.

### 2.5. The Intervention: Moshi Sleep App 

Moshi is a mobile audio-based application developed to help children that have trouble falling asleep to get to sleep more quickly (see Figure 2). Once the mobile app is downloaded on the user’s mobile phone (Apple or Android), it engages the user with sophisticated interactive guidance to choose from dozens of stories intended for preschool-aged children. The application is intended to be controlled by parents and when active in a story to have a locked screen, so that no light is emitted, and only the audio is projected from the mobile phone. Parents were advised to select one story per night to use with their child during the intervention period until their child fell asleep. A screen shot of the app is shown in Figure 2.

### 2.6. Measures 

We measured several sleep parameters at baseline and follow-up (see Table 2). We also screened children for potential behavioral diagnosis with the Copeland Symptom Checklist for Attention Deficit Disorders—Child and Adolescent Version [34].

#### 2.6.1. Pittsburgh Sleep Quality Index

The Pittsburgh Sleep Quality Index (PSQI) [35] is a validated self-reported questionnaire that assesses sleep quality and disturbance in adults over a 1-month timeframe. Survey items assess the following sleep areas: sleep quality, sleep latency, sleep duration, habitual sleep efficiency, sleep disturbances, use of sleeping medication, and daytime dysfunction. Scoring of responses on the PSQI is based on a 0 to 3 scale, where 3 reflects the higher levels of sleep disturbance and lower scores are better. PSQI global scores range from 0 to 21. A score on the PSQI of 5 or greater indicates poor sleep. The survey reported diagnostic sensitivity of 89.6% and a specificity of 86.5% (kappa = 0.75, *p* < 0.001) for identifying good and poor sleepers [35]. Parents completed self-reported PSQI at baseline (pre-intervention) and post-intervention.

#### 2.6.2. Child Sleep Habits Questionnaire (Abbreviated) 

The Short-Form Child Sleep Habits Questionnaire (SF-CSHQ) is a validated measure [36], that is widely accepted as a screening measure for parent-reported subjective assessment of 4–8-year-old children’s sleep [36]. The SF-CSHQ, consisting of 23 items, assesses children’s sleep habits and sleep-related problems, and uses the response options of ‘Usually’ (5–7 times/week), ‘Sometimes’ (2–4 times/week), or ‘Rarely’ (0–1 times/week). Higher scores indicate worse sleep behaviors or problems. Values over a score of 41 are considered to be clinically significant. The SF-CSHQ measures the following children’s sleep behavior domains: bedtime resistance, sleep onset, sleep duration, sleep anxiety, night wakings, parasomnias, sleep-disordered breathing, and daytime sleepiness. This study focused on sleep onset, duration and night wakings in the children. Parents reported on the SF-CSHQ at baseline and post-intervention. 

#### 2.6.3. Epworth Sleepiness Scale 

The Epworth Sleepiness (ESS) scale is a validated 8-item measure of daytime sleepiness based on the probability of falling asleep in a variety of situations [37]. Parents were asked to report their likelihood of nodding off or falling asleep in several situations, such as “sitting and reading” on a scale from 0 “Would never nod off” to 3 “High chance of nodding off.” A global ESS score is tabulated by summing the total responses. Higher scores indicate more sleepiness. Parents provided self-reports on the ESS at baseline and post-intervention.

#### 2.6.4. PROMIS Scale 

The Patient-Reported Outcomes Measurement Information System (PROMIS) [38,39] of Sleep Disturbance and Sleep-Related Impairment is a validated 8-item measure used to asses parents’ sleep-related impairment and sleep disturbance over the last 7 days. Items are rated on 5-point scales of intensity (*not at all, a little bit, somewhat, quite a bit, very much*), while a few items use a frequency scale (*never*, *rarely*, *sometimes*, *often, always*) and one item assesses overall sleep quality (*very poor*, *poor*, *fair*, *good*, *very good*). The PROMIS scale (lower scores reflect better sleep quality) was used at baseline and post-intervention to measure parent-reported sleep quality.

### 2.7. Statistical Analysis

Participant characteristics were described using means and standard errors (SEs) for continuous variables and using counts and percentages for categorical variables. Sleep measures were evaluated in child only, parent only, and child–parent dyad groups. The sleep measure for child–parent dyads was calculated as the sum of the child sleep measure and the parent sleep measure, and these measures were rescaled with a standard deviation of 1. Pre- and post-intervention measures were compared using the Wilcoxon signed-rank test. All statistical analyses were performed using SPSS software and R version 3.6.1 (http://www.r-project.org, accessed on 1 June 2021).

## 3. Results

### Inferential Statistics Findings 

Table 3 presents the pre- and post-intervention measures for the child group, parent group, and child–parent dyad. The average post-intervention SF-CSHQ measure was lower than that of the pre-intervention measures (mean: 2.32 vs. 2.96; *p* = 0.01), indicating significant parent-reported sleep improvement in children when using the Moshi app. While the changes in parent-reported sleep were in a favorable direction post-intervention (suggesting improvements in sleep), the differences in ESS, PSQI, sleep duration, and sleep efficiency were non-significant. For child–parent dyad comparisons, no significant findings were identified pre- and post-intervention. The average use time of the app was 15 min per night. The study results show improvements in sleep health pre- and post-intervention for all children enrolled in the pilot, and improvements in mothers’ daytime sleepiness.

## 4. Discussion

An audio-based application that improves sleep habits in children and parents could be a viable solution for improving sleep health for the family unit, especially among young children. This appears to be the first pilot of its kind to investigate the potential of an audio-based mobile sleep aid to improve sleep health in a racially diverse parent and child dyad sample [40]. Prior studies evaluating parents’ perceptions of their child’s engagement with the Calm app among children and adolescents younger than 18 years old reported that it was mostly used to improve sleep health (76%), stress (32%), depression or anxiety (28%), or overall health (14%). Sleep stories (95%), music and soundscapes (67%), meditations (65%) and breathing exercises (65%) were the most common components of the app used by children that parents found helpful for their child’s sleep [41].

Improvements in sleep health were reported for all children pre- and post-intervention. The results are promising, with parents who benefitted from the app reporting improvements in parents’ daytime sleepiness. This supports prior studies of parents with toddlers that reported daytime fatigue or tiredness, due to poor children’s sleep health at night, which has even been associated with affecting mothers’ mental health [42,43]. These studies are promising and suggest that strategies that improve sleep health for both mothers and children are critical for disrupting the loop of children’s sleep and maternal daytime fatigue in low-income families. Our pilot study reported improvements in sleep and wellbeing for both mother and child, which could improve the health of the family unit.

One strength of the study is its use among parent–child dyads in their natural home environment, even during the height of the pandemic in New York City. The ability to easily integrate the use of a mobile application into bedtime routines during a disruptive timeframe (including virtual schooling, childcare, working remotely, and potentially food, housing and/or financial insecurity) during the pandemic speaks to the utility of the application. Additionally, the affordable and accessible qualities and the ease-of-use of the mobile application in a racially diverse sample mean that it should be considered highly as a viable sleep aid solution, especially given the sleep health disparities affecting children of color [44]. This study’s strength also lies in the population that participated in the pilot, which included a racially diverse sample. Although the parents’ average age was similar to the larger United States population, our sample included a larger Black sample of parents who were educated with a Bachelor’s degree or higher, compared to the larger comparison parent population that was predominantly White with an average education lower than a Bachelor’s degree. This is a strength to our study that could contribute to addressing pediatric sleep health disparities among Black children who report disproportionate rates of shorter sleep duration, sleep quality and bedtime routines [44].

This pilot study is promising as a sleep intervention for parents and children. Mobile technologies are increasingly low-cost and ubiquitous. These tools hold promise for delivering a soothing bedtime routine, such as relaxing stories and music. Digital tools criticized for adversely affecting sleep health in children are based on the use of screen-based media consumption [12]. However, the utility of the mobile phone as a sleep aid is novel. The way in which the mobile device is used, i.e., as an audio-based aid (with the screen face down) means that it holds potential to be a helpful sleep aid solution. Additionally, audio-based bedtime stories may help with literacy, language acquisition, and cognitive and behavioral development. 

### Limitations

Our small sample size does not allow for the findings to be generalizable to a larger population. The study design in this study also does not allow us to determine efficacy. The within-subject design introduces bias, and study participants would need to be enrolled for a longer study period. The lack of a control group, wait list or educational control is a limitation as well. This study also did not take into account parents’ level of physical activity, BMI, sleep apnea, or pre-existing conditions, which is a study limitation, as this may confound the results. Significant findings in sleep health domains could serve as preliminary evidence for a future larger trial, powered to determine efficacy in a family-based sleep study. The accessibility of the mobile sleep aid on Android and IOS devices could hold public health significance to improve sleep health in children who are experiencing symptoms of ADHD or ASD, and improving sleep health and quality of life dynamics for the family unit. Improvements in sleep onset and night wakings could reduce parent burden during bedtime, a challenging timeframe that could be dreadful for parents. Improvements in parents’ sleep could contribute to better mood and household dynamic. 

Another limitation may be that audio-only stories may have a lesser impact on the cognitive and emotional development of children than physical books. Prior research found that reading books to children at bedtime was significantly associated with longer sleep duration in a study of Latino preschoolers [45]. However, in a study that compared brain activation between audio-only and text-based narratives, researchers found that the same parts of the brain were activated with both audio-only and reading books [46]. This suggests that the emotional and mental/cognitive benefits might be the same. 

## 5. Conclusions

Parent–child dyads reported better sleep health for children, suggesting a trend in sleep health for children and in sleep health for parents based on the preliminary evidence of the effects that were measured. The Moshi app is a promising intervention that could be incorporated into bedtime routines of school-aged children to promote healthy sleep. The audio-based solution induced better sleep in children, which may be a result of calming and relaxing feelings; however, additional research in this area is warranted as these factors have not been assessed in this study. Extending the utility of a sleep-based aid to a mobile phone device makes the idea contemporary. Mobile health applications, such as Moshi, enable smart phones to serve as digital change agents that are accessible and promising.

## Figures and Tables

**Figure 1 ijerph-19-09416-f001:**
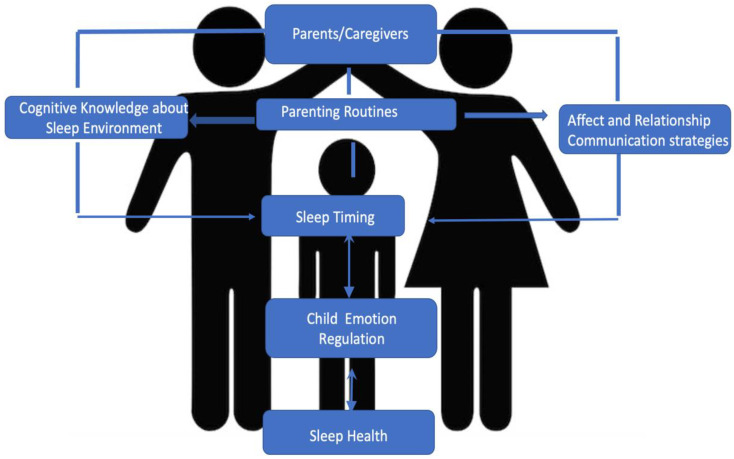
Theoretical framework guiding parent–child dyad sleep health in the family unit.

**Figure 2 ijerph-19-09416-f002:**
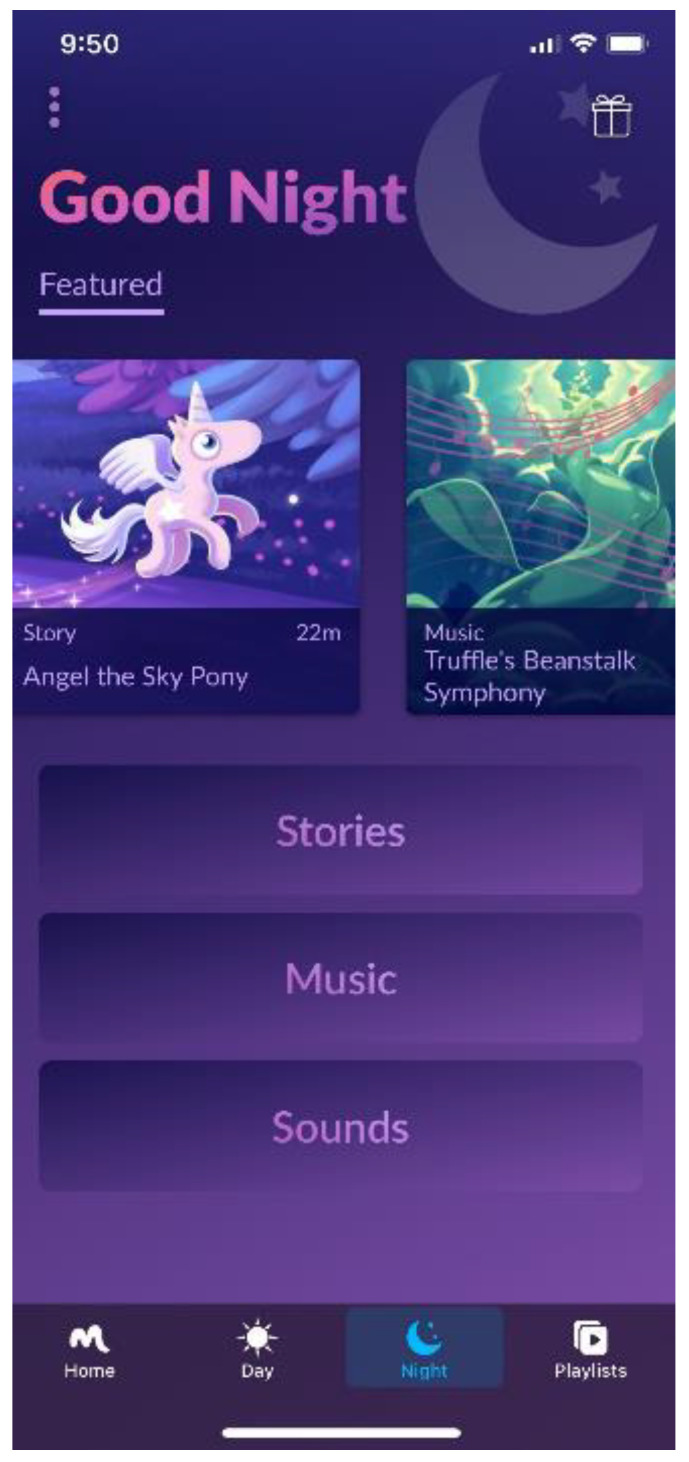
Screenshot of the Moshi audio bedtime stories mobile application.

**Table 1 ijerph-19-09416-t001:** Descriptive Statistics.

	**Total (*n* = 20)**
	Mean (SD)/N (%)
**Children**	
Age	4.29 (1.23)
Male	15 (62.5%)
**Parents**	
Age	35.90 (5.83)
Female	16 (84.0%)
Married	19 (76.0%)
Employed	20 (100.0%)
Race	
White	6 (24.0%)
Black	12 (48.0%)
Other	7 (28.0%)
Bachelor’s degree or higher	16 (64.0%)

Note: Abbreviations: SD—standard deviation.

**Table 2 ijerph-19-09416-t002:** Schedule of survey measures administered.

	Screening	Baseline	Follow Up
Copeland Symptom Checklist for Attention Deficit Disorders—Child and Adolescent Version	X		
Demographic		X	
PSQI		X	X
PROMIS		X	X
ESS		X	X
SF-CSHQ		X	X

Note: abbreviations: PSQI = Pittsburgh Sleep Quality Index; ESS = Epworth Sleepiness Scale; PROMIS = Patient-Reported Outcomes Measurement Information System; SF-CSHQ = Children’s Sleep Habits Questionnaire.

**Table 3 ijerph-19-09416-t003:** Pre-intervention vs. post-intervention.

	**Pre-Measure**	**Post-Measure**	***p*-Value**
	Mean (SE)	Mean (SE)	
**Child only (*n* = 20)**			
SF-CSHQ	2.96 (0.55)	2.32 (0.55)	0.010
**Parent only (*n* = 20)**			
PROMIS	17.80 (5.96)	20.04 (2.91)	0.445
ESS	7.00 (4.91)	6.27 (4.87)	0.782
PSQI	5.28 (3.29)	6.00 (3.44)	0.226
Sleep duration (self-report)	6.19 (1.05)	6.46 (1.13)	0.164
Sleep efficiency (self-report)	0.89 (0.12)	0.89 (0.14)	0.776
**Child–parent dyad (*n* = 20)**			
SF-CSHQ + PROMIS	8.35 (1.08)	8.08 (1.58)	0.697
SF-CSHQ + ESS	6.79 (1.25)	5.54 (1.61)	0.052
SF-CSHQ + PSQI	7.08 (1.10)	6.02 (1.73)	0.065

Note: Abbreviations: PSQI = Pittsburgh Sleep Quality Index; ESS = Epworth Sleepiness Scale; PROMIS = Patient-Reported Outcomes Measurement Information System; SF-CSHQ = Children’s Sleep Habits Questionnaire. SE = Standard Error.

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
