# Peer review of "Out Like a Light: Feasibility and Acceptability Study of an Audio-Based Sleep Aide for Improving Parent–Child Sleep Health"

_ijerph, 2022, doi:10.3390/ijerph19159416_

Round 1

Reviewer 1 Report

Thank you for significantly improving this manuscript.

It is however a pity that you reported after such small numbers and duration, and that your sample was so small, subjectively recruited, and with no explicit acceptance rate.  These factors seriously undermine its value as a representative feasibility study.

Line 291 et seq. - this is a bold and unprovable statement - you should at minimum moderate it to 'formally published study', and preferably precede with 'appears to be'.

Lines 354-357 - this is a very complex and unclear sentence.

Reviewer 2 Report

Dear authors, thank you for your careful review and responses to my comments. I believe the conclusion paragraph is still misleading. As worded,  it implies that you have established the evidence for this in your study. That is incompatible with your study, especially as a feasibility study. The parent child dyads reported improvements, but significance is also misleading here. I suggest editing this paragraph to better fit the rest of the paper, which reports a promising intervention/ap, and improvements in what you measured as preliminary evidence of an effect, not established evidence of efficacy or effectiveness. 

"Parent-child dyads reported significant improvements in sleep health for children and suggests improvements in sleep health for parents. The Moshi app could be incorporated into bedtime routines of school-aged children to promote healthy sleep. Evidence  on audio-based solutions to induce calming, relaxing feelings to induce sleep has been established. "

Reviewer 3 Report

Review Chung et al, 2022, Int J Env Res and Public Health

The manuscript “Out like a light: feasibility and acceptability study of an audio-based sleep aide for improving parent-child sleep health” by Chung et al explored if sleep health can be improved by a smartphone-based app that can tell bedtime-stories to children of the age 3-8 years old. Both the sleep of the children as well as the sleep of the parents was investigated.

The study was performed fine and the results are promising, although not very significant. Yet, I deem this study to be publishable as is, with a few minor comments that could improve the study.

I think the conclusions in the Abstract are too strongly stated (lines 31-32). No small change has been reported for parents, just a trend can be seen, but the significance levels were not appropriate to call this a change. Also in the Conclusions, the sentence starting with “Evidence…” (lines 361-363) is too strongly stated. In this study sleepiness has been investigated. It can not be stated with the methods used that the audio-based solution actually induce calming and relaxing feelings. It can be said that it induced better sleep in children, which may be caused by calming and relaxing feelings, but these factors have not been assessed in this study.

I think the app-based bedtime stories should also be compared to parents reading bedtime stories to children. A possible reference is Brown et al, 2016, Clin Pediatr (Phila), https://doi.org/10.1177/0009922815593907. I think that the Brown et al study may strengthen the results in this study, because sleep improves in children after reading to them.

In the introduction (lines 101-102) it states that the lasting effects of … has not been reported, suggesting that in this study the lasting effects ae reported. In my opinion, a study of five days does not report lasting effects. I think this is too strongly stated.

In figure 1 the vertical arrows can barely be seen, as well as the arrows going ‘around’ (not sure where these point to). Also this figure is confusing to me, as in the text multiple factors are mentioned that I do not see in this figure. Either the figure is not explained well in the study, or the figure does not describe the text well. As it is now, I find it very confusing. Please describe the figure in the text, and then perhaps elaborate on this.

In the Materials en Methods section, lines 167-168 show a line of Abbreviations, but I can not find where this line belongs to.

The study procedure states that each participant received 5 days of normal sleep procedures and 5 days of exposure to the mobile app. Each contains 3 week days and 2 weekend days. How are these planned: first normal sleep 3 week days and 2 weekend days, then 2 days “off”(?) and then 3 week days with the app assistance and 2 weekend days? Then the 2 days “off” are also ‘normal’? Can you state how these two periods are chosen? Is this the same for all participants? And if not, does this make a difference in the results?

On page 8, lines 283, 301 and 307, the word ‘mother’ was chosen above the word ‘parent’ that is used in the rest of the study. Have the ‘fathers’ been excluded for this part? Or were all parents mothers?

Author Response

This manuscript is a resubmission of an earlier submission. The following is a list of the peer review reports and author responses from that submission.

Round 1

Reviewer 1 Report

This is a potentially interesting study.  However, apart from the sample size being small, there are serious unanswered questions about its  representativeness

- how were the recruitment locations selected?

- what was the acceptance rate?

- were the participants socially/societally representative?

- how were parents invited/tempted to participate?

- were all parents invited, then their children's ASD/ADHD identified, or were locations servicing ASD/ADHD children selected?

- does the education/age/ethnicity profile of the sample match that of the overall parental population?

- within the app, what selection is there of stories - age, interest, culture, etc.?

Reviewer 2 Report

Dear authors,

Overall, this paper has a fantastic introduction and background. You make a very compelling case for your study and app, but the limitations of the study, as currently reported, are considerable, and I believe it needs major revision at present. It is a fascinating paper and I think there is great potential for further development, however!

Comments/Queries:

Line 93: Surely, this study is meant to determine if the app is promising? What is this based on in this sentence? Has this been pilot tested, or has there been a feasibility study?  See my comments about line 146 as well.

Line 95: Just a comment, this is a very well written and considered introduction and background. Well done considering the social-environmental factors and the effect of family routines on sleep. Mealtimes are a particularly good addition to this section. 

line 103: The positive effect on language and verbal test scores may also be mediated by sleep, given the memory consolidation and other effects of sleep on cognition and memory. 

Line 107: Great point! This is an important gap to highlight. Dyadic sleep interventions have great potential for improving well-being in families and communities (as has been seen in dyadic sleep interventions for adult caregivers of older parents, among others).

Line 115: Is the citation for this the same as for the next line?

Lines 118-127: Why may this be? What may underlie sleep disturbances in this group?

Line 146: This needs more explanation, a citation, or explanation. What techniques, and what does that mean? How evidence-based? Was it or has it been pilot tested? Who was it developed by, or where? This line seems to be dropped in here, but it requires more explanation.

Line 152 and that section: excellent!

Lines 172-179: These are good hypotheses but incompatible with a pilot study. A pilot study is typically not powered to provide evidence of effectiveness. These are hypotheses compatible with a different kind of study design. It would not be appropriate to call it a pilot study because it may have a small sample size. Why was a pilot study conducted, rather than a full trial of some kind?

Line 179: As you had only 5 children with ASD or ADHD, you do not have enough of a sample of children with ADHD and ASD to determine that the disorders moderated the effects, rather than individual differences outside of them. There are similarities but also vast differences between the temperaments, habits, sensory processing, regulation self- calming abilities between children with ADHD and the autism spectrum.  

 Also, moderate how? How did you expect the disorders to moderate what effects?

Line 182: Again, a pilot study does not provide evidence of efficacy. It provides information about the feasibility of recruitment, study procedures, participant or provider willingness or equipoise, etc. You could get early indications or “signals” of an effect or safety, but you can’t determine efficacy with a pilot study.

Lines 187 to 195: I doubt that you have a study design that allows you to make determinations of efficacy. A within subject design can be biased by the effects of time or carry over effects if parents have any expectancy bias, knowing they will participate in an intervention soon. I know there are practical advantages to this kind of design, but the disadvantages are also notable for an investigation of efficacy and with a short time frame like yours. With only 1 weekend of 2 weekend, you can’t really accommodate differences in family or child routines that are habitual and influence sleep. Similarly, only 3 weekday nights is a very short time period, especially with a within-subjects design, and no time for “accommodation” to the change in routine that may come with the intervention and mask effects of the intervention.

Line 194: I wonder again if this small sample of 5 child-parent dyads with two disorders known for their very variable presentations is sufficient for you to assess their moderating role on the intervention. 

Lines 203-208: What about parents’ level of physical activity or sedentariness (bidirectional relationship with sleep and effect on circadian rhythms)? Alcohol or tobacco, drugs? Also, BMI/obesity or sleep apnea, chronicity, or pre-existing sleep disorders? Were these screened for, excluded, or considered? Any of these could confound the results.

Line 218: The Copeland is not intended to screen for ASD. How is this used to screen for ASD here? Are parents reporting the child has been diagnosed with ASD, or they suspect? How does ASD get determined, and is it a reliable enough way for it to be determined? 

Line 224: Does this mean 5 children with ASD and ADHD total? 

Lines 228: There would have been the possibility of using actigraphy for the parents (in a future study?). Of course, actigraphy has its drawbacks as well.

Lines 238-239: If you say something has been designed with evidence-based techniques, you also need to say what they are and what the evidence is or cite a previously published peer-reviewed source that contains it. What do you mean by evidence-based techniques? That could mean anything from a case-series to a large double blinded RCT or an observational study. This is unclear, and you have repeated it twice in a few pages. It is an important point you are making but you are not providing any evidence for it or explanation of it. 

Lines 237-246: There is very little information about the app itself. Do parents download it onto their phone? Is it collecting data on their phone? Who developed it? Has it been tested? Safety or feasibility information? How much can individual users modify the app or how it works, or is it pre-set and cannot be modified (brings up questions about fidelity in the study).

Lines 237-246: This information is needed: How did the app work? How long was the pink noise, vs white noise, vs other components? What kind of music? What volume, or was that up to the user?  There is very little information in the paper about how the intervention actually proceeds or works. How long is the app used for? Is it until the child falls asleep, or for how many minutes?  How long before sleep, or after getting in bed is it used? Were there any instructions for parents in terms of bath times, mealtimes, activity before bed, lighting, etc? Without this information, it is very hard for the reader to make a sensible determination about the utility, feasibility, therapeutic or mechanistic reasoning behind of the app.

Line 250: You took baseline PSQIs. Was there a cutoff? Were all parents/children eligible, even if parents had high PSQI scores, for example? If so, was that accounted for in your analyses?

Line 266: would your short intervention time allow for changes the PSQI or SF-CSHQ could detect in such a short time?

Lines 337-339: What were the baseline scores on the SF-CSHQ for children with ADD/ADHD and separately, children without them (table 4)? This is telling the reader where they ended up, but one needs to know where they started as well.

What differences would be considered clinically significant? You have a pilot study, so you can’t really make claims for efficacy, and a small sample size, making this complex as well.

Line 358: This wording is misleading. There were statistically significant differences for very small changes- this is very different than there being significant improvements. Are any of these differences clinically significant on those measures? Those were very small differences, and do not strike the reader as being significant improvements, especially given the small sample size and very short length of the study. 

Line 378: Please see my comments for lines 237+

Line 382: What db and for what duration did the Moshi app deliver the pink or other noise? How was this monitored? If on the app, how was the data collected and analyzed?

Lines 400-401: Is this how the Moshi was used? What about light emitting from it if not? 

Lines 405: See my previous comments about your sample size. It isn’t just generalizability. The lack of a control group, even a wait list or educational control makes the study design a limitation as well.

Lines 429-420: Did parents stay in the room with the child to use the ap (presumably the children didn’t have smartphones of their own at that age?), or was it left in their room and the parents later retrieved their devices? That may be a benefit of the old-fashioned cassettes- they don’t emit light, and a parent doesn’t have to retrieve them.  What happens if a parent gets a phone call, text, or social media notification on their phone while using Moshi? Would it interrupt the program? 

Line 434: This is not a statement you can make given the limitations of the study and the fact that it is a pilot. It would be wonderful if you find, with a larger sample and more rigorous design, that you have an effective intervention/device/app, and that it could be used in the way your conclusion states, but it is a bit misleading to state that at present in the conclusion, unfortunately.

Reviewer 3 Report

Alicia et al. presented the efficacy of an audio-based sleep aid named Moshi. Although the study is novel, methodologically the study needs more improvement. Here are some comments:

1) The study had overall two objectives: 1) Showing the efficacy of the intervention which was performed by 5 nights with the dyad. and 5 nights without the dyad. 2) Showing the efficacy of the dyad in ADHD/ASD vs non-ADHD/ASD parent-child. However, it wasn't clear in the overall presentation of the study. I would suggest making a clear distinction between these two.

3) Given only 5 participants in one group, this finding has severely limited validity. I am not sure even if this should be included in the manuscript. 

3) Overall, the study cohort was very small when performing the efficacy of an app. The statistical power of the study should be calculated.

4) Also, the duration of only 5 nights needs more rationale. Why the 5 nights were chosen? Also, how the intervention is generalized in terms participants' overall health? How it is accounted for the fact that participants may have mental or physical stress during those 5 nights?  

5) Introduction is very long. I would suggest cutting it in half.